# Experiences of Carers and People with Dementia from Ethnic Minority Groups Managing Eating and Drinking at Home in the United Kingdom

**DOI:** 10.3390/nu14122395

**Published:** 2022-06-09

**Authors:** Pushpa Nair, Yolanda Barrado-Martín, Kanthee Anantapong, Kirsten Moore, Christina Smith, Elizabeth Sampson, Jill Manthorpe, Kate Walters, Nathan Davies

**Affiliations:** 1Research Department of Primary Care and Population Health, University College London (Medical School), Upper Third Floor, Royal Free Hospital, Rowland Hill Street, London NW3 2PF, UK; y.barrado@ucl.ac.uk (Y.B.-M.); k.walters@ucl.ac.uk (K.W.); n.m.davies@ucl.ac.uk (N.D.); 2Marie Curie Palliative Care Research Department, Division of Psychiatry, University College London, 6th Floor, Wing A, Maple House, 149 Tottenham Court Road, London W1T 7NF, UK; kanthee.anantapong.18@ucl.ac.uk (K.A.); k.moore@nari.edu.au (K.M.); e.sampson@ucl.ac.uk (E.S.); 3Department of Psychiatry, Faculty of Medicine, Prince of Songkla University, Hat Yai, Songkhla 90110, Thailand; 4Melbourne Ageing Research Collaboration, National Ageing Research Institute, 34-54 Poplar Road, Royal Melbourne Hospital, Victoria 3050, Australia; 5Language and Cognition, Division of Psychology and Language Sciences, University College London, Chandler House, 2 Wakefield Street, London WC1N 1PF, UK; christina.smith@ucl.ac.uk; 6Department of Psychological Medicine, Royal London Hospital, East London NHS Foundation Trust, Whitechapel, London E1 1BB, UK; 7NIHR Policy Research Unit in Health & Social Care Workforce and NIHR Applied Research Collaborative (ARC) South London, King’s College London, Strand, London WC2 4LL, UK; jill.manthorpe@kcl.ac.uk

**Keywords:** nutrition, hydration, dementia, eating, drinking, food, ethnic minorities, culture

## Abstract

Eating and drinking difficulties, such as loss of appetite and swallowing problems, are common in dementia, but little is known about the experiences of ethnic minority groups who are managing these difficulties at home. The purpose of our study was to explore the meaning of food, the impact of dementia on eating and drinking, and carers’ experiences of support. We undertook semi-structured interviews with 17 carers and people with dementia from ethnic minority backgrounds living in England, using thematic analysis to analyse the data. Food/drink had strong links to identity, culture and emotions. Providing culturally familiar foods, celebrating traditional festivals and supporting previous food-related roles promoted reminiscence, which encouraged the people living with dementia to eat and drink, as did social interactions, although these could lead to distress in those with more advanced dementia. Food choices were also influenced by carer strain, generational differences and the impact of health conditions. Despite a strong sense of duty to care for relatives at home, there was low awareness of community support services. The carers expressed a need for culturally tailored support for managing dementia-related eating and drinking difficulties at home. Healthcare professionals must provide contextually relevant advice to carers, being mindful of how cultural backgrounds can affect dietary choices.

## 1. Introduction

In 2019, more than 920,000 people were living with dementia in the United Kingdom (UK), with an estimated 25,000 from ethnic minority groups [1]. There is a higher prevalence of dementia in ethnic minority groups [2], with higher rates of earlier-onset dementia [3] and evidence of later presentation to dementia services, with consequent delayed diagnoses [4].

Dementia is a progressive neurological condition that can affect memory, problem-solving, language and behaviour, with the most common types being Alzheimer’s disease, vascular dementia, frontotemporal dementia and dementia with Lewy bodies [5]. Whilst the progression of dementia is highly variable [6], eating and drinking difficulties are common towards the later stages, with over 80% of people living with dementia encountering at least one eating or drinking-related difficulty [7]. Difficulties include loss of appetite, reduced consumption, holding food in the mouth, over-chewing, not recognising food or utensils [8] and swallowing difficulties, which can lead to dehydration, malnutrition and aspiration [9]. These may be further exacerbated by behavioural changes, such as apathy and restlessness [10]. Even in the earlier stages of dementia, memory problems can affect meal preparation and lead to changes in food preferences [11]. Managing eating and drinking difficulties at home is often challenging and stressful for family carers [12,13]. Most people with dementia in the United Kingdom, and especially those from ethnic minority groups, are supported at home [14].

People living with dementia who develop eating and drinking difficulties might be supported by a number of strategies, ranging from simple encouragement to food-texture modification, fluid thickeners, specially adapted utensils and hand feeding [15,16]. However, the specific guidelines [17,18] for managing nutrition and hydration in dementia can be unclear and often lack culturally specific advice.

Food and drink are more than physical sustenance; they are intrinsically tied up with an individual’s personality, cultural identity, social practices [19] and religious beliefs [20], in addition to general enjoyment and quality of life. For people living with dementia, food habits may be significantly shaped by personal upbringings, cultural backgrounds and, for some, migration journeys. The loss of more recent memories may further affect food preferences and the ability to articulate them [21]. Research into eating and drinking in dementia has predominantly focused on long-term care settings, such as care homes. Studies specifically exploring the experiences of people with dementia from ethnic minorities in care homes [19,22] have found that the provision of culturally familiar foods increased joy, fostered a sense of cultural identity and dignity, improved appetite and nutritional intake, promoted reminiscence and generally improved the care quality. However, the meaning and role of food and drink for people with dementia living at home have only been explored in a few studies. These studies highlight the importance of cultural foods and mealtimes for identity, wellbeing and social connectedness [23,24,25]. There remains a gap in the literature exploring the relational and emotional aspects of eating and the practices related to this, and their importance in structuring the everyday experiences of care for people living with dementia. The lack of studies in this area, together with the growing ethnic diversity of the UK’s population and evidence of disparities in dementia care for ethnic minority groups [26,27], highlight the importance of this research.

This study aimed to explore the experiences of people living with dementia and carers from ethnic minority groups in England in relation to managing eating and drinking at home. We interviewed carers supporting people with moderate–advanced dementia, as well as a smaller sample of people with early stages of dementia, to include a range of experiences. We explored views related to the meaning and cultural importance of food and drink, the impact of dementia on eating and drinking habits, carers’ experiences of support and unmet needs. We found that cultural backgrounds were influential on attitudes and experiences related to eating, drinking and care.

## 2. Materials and Methods

This was a qualitative study using semi-structured interviews with people living with dementia and family carers. Our objectives were twofold: (1) to explore the cultural importance and meaning of food and drink for people living with dementia and their carers in ethnic minority communities; and (2) to explore the role of carers in supporting eating and drinking in dementia at home in ethnic minority communities.

### 2.1. Participants

We recruited participants from a variety of sources over a period of nine months in 2019/2020, including GP practices, memory services, social media (e.g., Twitter, using departmental accounts), previous studies, online dementia research and carer support websites (such as Join Dementia Research (JDR)), local carer organisations and dementia services. Participants were purposively recruited from neighbourhoods within and around Greater London to sample different ethnicities, socioeconomic backgrounds and levels of access to health and dementia care services.

GP practices and memory clinics sent out postal invitation letters and study information leaflets (with prepaid postal reply slips) to potential participants who met the inclusion criteria listed below in Table 1. Participants identified via social media or online research websites (e.g., Twitter, JDR) were sent postal invites (with prepaid postal reply slips) or email invitations with study leaflets attached (according to the participant’s preference). Participants identified through dementia hubs or carer organisations were first approached by members of that organisation, and then contacted by the research group as stated above. Interested participants were then telephone-screened by the first and second authors to confirm that they met the inclusion criteria and to give them an opportunity to ask any questions.

### 2.2. Inclusion and Exclusion Criteria

Inclusion and exclusion criteria for this study are listed in Table 1 below.

Ethnicity was assessed during telephone screening (self-reported by participants), as well as in a demographic questionnaire completed at the end of interviews (whereby participants either ticked the corresponding ethnicity box that they identified with, set out in accordance with British census criteria [28], or wrote their ethnicity down in the alternative white-space box provided).

### 2.3. Data Collection

Data were collected through individual semi-structured interviews, most of which were conducted face to face in participants’ homes (or another location if requested), and written informed consent was obtained prior to interviews. Interpreters were available if needed, but no participants requested this. Interviews were conducted by PN (*n* = 16, an academic GP) and YBM (*n* = 1, a qualitative health researcher). Because of the impact of the COVID-19 pandemic, three interviews were conducted remotely (two by telephone, one using Microsoft Teams). Interviews lasted approximately one hour on average (range: 35 to 111 min) and were audio-recorded using a Dictaphone. The interview topic guide was developed with the input of the whole research team, including our patient and public involvement (PPI) representatives, and was modified as interviews progressed. The topic guide explored the following subjects: reported changes in eating, drinking and food preparation since dementia started; carers’ experiences of managing/preparing food and drink within different cultural contexts; cultural practices and meanings related to food and drink; carers’ experience of support; and identifying unmet needs. At the end of each interview, we collected demographic data and participants were provided with a £20 voucher. Audio-recorded interview data were transcribed verbatim by an external transcription service, anonymised and verified for accuracy by the interviewer. Sample size was guided by the concept of information power [29].

### 2.4. Data Analysis

Thematic analysis was used to analyse the data from a constructivist perspective, which emphasises the importance of multiple viewpoints, contexts and values [30]. The team included two academic GPs (PN, KW), two qualitative health researchers (YBM, ND), two psychiatrists (KA, ES), two social gerontologists (KM, JM) and a speech and language therapist (CS). All transcripts were read by PN and YBM. PN then developed an initial list of codes, which was further refined iteratively throughout analysis by reflecting on the transcripts and through discussion with the team about ideas. PN coded transcripts line by line using NVivo 11 [31], according to the initial thematic framework, which was refined further as transcripts were coded, and with the input of all authors. PN met regularly with YBM and ND to discuss, agree on and refine emerging analytical themes, with the input of KW. The final themes, which highlighted relationships between ethnicity, cultural backgrounds, eating, drinking and care, were discussed and agreed upon by all authors.

## 3. Results

We recruited 17 participants. Of these, 5 were people with early-stage dementia (three female, two male), and 12 were current or former carers of people in the later stages of dementia (eight female, four male). Of the people living with dementia that were interviewed, the mean age of the participants was 75.6 years (range: 57–87 years), and of the carers interviewed, the mean age was 55.25 years (range: 29–81 years). See Table 2 for full demographic details.

### 3.1. Themes

The findings from our analysis centred around four main themes: (1) cultural expectations around care; (2) food as an expression of culture, identity and relationships; (3) barriers to engaging in cultural food practices; and (4) the need for culturally sensitive services. Illustrative quotes for each theme can be found in Table 3 below.

#### 3.1.1. Cultural Expectations around Care

The cultural expectations around care emerged as a common topic during the interviews. Most (but not all) carers managing the practicalities of cooking for their relatives with dementia were female, particularly amongst South Asian participants, where it was commonly perceived to be a cultural duty of care, even if the main listed carer was male. This included not only daughters and daughters-in-law, but also wives, who, in some cases, were quite elderly themselves and struggled with the added pressure:

*‘But the South Asian community, when I consider my father-in-law, he always thinks that it’s his wife’s duty [to undertake all caring duties, including preparing meals] and he didn’t need any carers [care workers] to come in and go out, it all has to be done by his wife.*’ (Former carer of parent-in-law, male, South Asian ethnicity, C11.)


*‘It was unfair on my wife, she had to bear that burden. And I used to try and make my mum understand. Mum, be appreciative of what the fact is, that my wife is putting in this effort, making it [meals] for you. God forbid if you were alone with me, how would you have survived?’*
(Carer of parent, male, South Asian ethnicity, C5.)

Most carers, across all cultures, said it was a cultural expectation in their communities to care for their older people at home, rather than consider residential care:


*‘In the Caribbean culture, there’s pressure to keep your parents at home’*
(carer of parent, female, Black ethnicity, C12.)


*‘We’re Asian, so we have to keep our mum at home’*
(carer of parent-in-law, female, South Asian ethnicity, C6.)

Some expressed concerns that the food in care homes would not be culturally appropriate, cultural festivals would not be celebrated or that language barriers would prevent their relatives from being able to express their food preferences:


*‘I think we’d have 24-h care with carers (care workers) coming in [rather than a care home]…because I think mum would lose the will to live because the food is not going to be culture-specific. It’s not like school, at least schools you take in celebrations, so if there’s Eid or Ramadan you include it. But in care homes, I don’t think they do stuff like that. So, if they did, if they made it more multicultural, it would be more bearable.’*
(Carer of parent, female, Black ethnicity, C8.)

This carer also felt that culturally inappropriate food in care homes could exacerbate distress:


*‘Talking about food, my mum’s carer (care worker), she mentioned about this chap. He was, I think he was Greek Cypriot. And she went to visit somebody else in this care home, and they had given him some sausage and mash or something. He looked at it and he threw it across the room, and, of course, they’re going to say that he’s aggressive. He wasn’t aggressive, he was confused, he was frustrated and the food wasn’t right……He probably didn’t recognise that it was food. He wasn’t being horrible, he was just was like, what the hell are you giving me?’*
 (Carer of parent, female, Black ethnicity, C8.)

Some carers reported that their relatives with dementia had expressed a wish to return to their home country in their old age, and that they felt that it was their duty to help support them to fulfil this wish. These carers felt that their relatives with dementia might receive care back home that would be more culturally acceptable to them, which would encourage them to eat and drink. In our sample, a South Asian carer had relocated his relative with dementia back to India and reported being happy with the culturally appropriate care that they were now receiving. Three Black Caribbean carers mentioned that they were either thinking about doing this for their relatives with dementia in the future, or expressed regret at not having done this when their relative with dementia was alive, as they strongly believed it would have encouraged them to eat and drink more:

*‘I used to work for the [name of charity] and I see people brought their family from Asia into England at 80, they come from a village and in no time they deteriorated. They deteriorated because they moved them out of their cultural place and they come here, because families think, they’re going to have a better lifestyle in terms of food, drink, whatever and a better environment. But you take that person out of that culture and bring them here, this place is isolating, but if you live in a small village in Kolkata or something it’s fantastic, do you know what I mean?*’(Former carer of parent-in-law, male, South Asian ethnicity, C11.)

#### 3.1.2. Food as an Expression of Culture, Identity and Relationships

For many participants living with dementia, food and drink, and specifically practices related to cooking and eating, were an important source of identity. Food and drink were also linked to emotions of love and loss in carers, associated with the dementia process and the ensuing role changes.

Identity, roles and emotions

Many carers reported that food-related practices, such as cooking and hosting, had represented a fundamental part of the person with dementia’s identity. This was especially true for women, who had commonly held significant food-related roles in earlier life, associated with providing food for their families as wives and mothers, and this was evident across all cultures:

*‘It’s more about being an Asian woman. Right from the early age, it’s like looking after the family. So, she’s always, whoever comes, a guest in the house, she will always offer them food.,,,,[..]……. Even now, today, the lady [cleaner] was saying, the new lady that’s she’s got. She said, are you still cooking? Oh yes, when I’m home I still cook. So she still thinks she does the cooking.*’(Carer of parent, female, South Asian ethnicity, C9.)

A sense of these identities and roles often persisted after the onset of dementia, even in those with advanced dementia, particularly amongst the South Asian participants. This took the form of upholding certain engrained etiquettes during mealtimes, such as wives only eating after their husbands had been served:


*‘She will not, by herself, will not come to the table…. And I’ll say come on, come to the table. But the one thing that she still does is she always looks at my dad to say, are you not joining us? Come on, sort of a thing, you know. It’s that her… Because she, being the woman, looked after the family for all her life.’*
(Carer of parent, female, South Asian ethnicity, C9.)


*‘He’s the main person, the head of the family. He’ll sit on one side in the bigger chair and he wants everyone to be with him [during meal-times]’.*
(Former carer of parent-in-law, male, South Asian ethnicity, C11.)

For some, cooking formed so much a part of the identity of the person living with dementia that changes in cooking style or a loss of interest in food and drink were sometimes the first signs of dementia noticed by family members:


*‘I start noticing that when she cooked the soup on a Saturday, that’s when it all came about, it didn’t taste the same. I remember saying, here, Mother, why this soup aren’t taste nothing. She used to get quite insulted about it, she used to get uptight….And then we started thinking, something’s wrong here.’*
(Former carer of parent-in-law, male, Black ethnicity, C4.)

For one participant with dementia, cooking was such an integral aspect of her identity that her grandson had created a cookbook of her recipes so that she could pass these on to future generations:


*Researcher: And what’s important to you about eating?*

*Participant: …Teaching them how to cook my way.*

*Researcher: Are there specific recipes that are important for you to teach?*

*Participant: Yes…For instance stuffed onions which takes a lot of time. I taught them but they love doing it so I don’t do it anymore, they do it. And things like that, yes.*
 (Person living with dementia, female, North African descent, PD1.)

It was also apparent in the interviews that, for many carers, feeding/eating practices were deeply emotional and were linked to exchanges and displays of love. Carers believed that it was their duty to put effort into mealtimes to encourage their loved ones to eat, and that this effort would be recognised as love by the person with dementia:


*‘And I’m preparing the food, getting home quickly, putting it on to the table, putting it in front of them. You forget, although you’ve done all of that, the person’s still there. In their mind, they would have seen you do all of that.’*
(Carer of parent, male, South Asian ethnicity, C2.)

For many, offering sweet foods was seen to invoke feelings of pleasure in the person with dementia. Whilst all carers acknowledged that sugary foods had limited nutritional value, they were reportedly used in moderation (and depending on existing health conditions) to display affection and care for the person with dementia, as well as to mitigate feelings of guilt associated with withholding such foods:


*‘So I used to, I’ve got a sweet tooth, so I used to sometimes say, I’m having a bit of chocolate or something, and I used to see her noticing me and I felt bad, because I didn’t want her to feel he’s not giving me these things. So I used to give her.’*
(Carer of parent, male, South Asian ethnicity, C5.)

In general, carers felt a sense of responsibility towards their relative with dementia’s eating and drinking, and, for some, this represented a sense of parent–child role reversal:


*‘To be honest with you, for me it was a huge learning curve going on and there was a role reversal going on, which I had heard about but never experienced.’*
(Carer of parent, male, South Asian ethnicity, C2.)

For some carers, witnessing their relative with dementia lose weight and interest in food constituted a form of grief and loss, especially if food had formed a large part of his/her identity prior to the onset of dementia:

*‘It was sad to see how this joyful woman who liked her food detested food*.’(Former carer of parent-in-law, male, Black ethnicity, C4.)

Some participants living with dementia also recognised that changes in their eating and drinking could have a negative emotional impact on their family members:

*‘I wouldn’t like what they [the hospital staff] give me to eat, would I? My daughter would have to bring things to me every day, and I thought she wouldn’t be able to do it, because she would break down. Because when I was in hospital and I wouldn’t eat it, I could see it on her face.*’(Person living with dementia, female, Black ethnicity, PD2.)

Cultural and religious beliefs

Most participants stated that culturally familiar foods were important to them. Carers reported that their relatives with dementia were more likely to show interest in eating if offered traditional foods during mealtimes, and they subsequently strived to obtain these:

*‘We insisted on getting things that she liked and things that were cultural to her.*’(Former carer of parent-in-law, male, Black ethnicity, C4.)

Most participants living with dementia corroborated this by reporting that they enjoyed eating foods in line with their cultural heritage, and that they would like to continue to be offered them as their dementia progresses:


*‘I was born in [North African Country] and I’m Jewish. And so there are specific Jewish foods and my family follow which I carry on [making] here.’*
(Person living with dementia, female, North African descent, PD1.)

*‘I think it’s quite difficult to go to a completely different place and have completely different food, in terms of if you’re Asian and then just English food. I think it’s important that they have things that they’re used to.*’(Person living with dementia, female, South Asian ethnicity, PD4.)

Our sample included participants from a range of religious backgrounds, including Christianity, Islam, Hinduism, Judaism and Jainism. For some of them, religious beliefs were integral to their dietary habits and preferences:


*‘We know for a fact that a lot of different people, with different religious beliefs will not eat certain foods, and so on. My mum is a prime example.’*
(Carer of parent, male, South Asian ethnicity, C5–Hindu vegetarian.)


*‘Yes, garlic, onion, like the cassava that grows under the ground, and the sweet potatoes, we don’t eat’*
 (Carer of spouse, female, South Asian ethnicity, C010–Jain religion.)

Alternative health beliefs, such as Ayurveda, influenced dietary choices for a minority of participants:

*‘She went to this Ayurveda doctor, and it was herbal medication he was giving, and changing your diet a bit, and that fixed the problem for her. So that was brilliant.*’(Carer of parent, male, South Asian ethnicity, C5.)

However, the person living with dementia could sometimes forget previously held values and beliefs surrounding food, which could be difficult for carers to manage:


*‘She’s always been vegetarian, yes. So that bit… Because one day, at Dad’s house, by mistake, she did help herself from the fridge, which…...She helped herself with a non-veg dish and she didn’t realise it. It was only later on that Dad realised that she had eaten it.’*
(Carer of parent, female, South Asian ethnicity, C9.)

Food as a connection to the past and others: reminiscence and social interaction

Many of the carers believed that certain cultural foods could evoke positive associations or memories of homelands, childhoods and family life for their relatives with dementia, and subsequently felt that these foods encouraged them to eat and to take a greater interest in mealtimes. Some mentioned that their relatives with dementia requested specific recipes from their childhood:


*‘And when he was very frail there was a time when he did crave the things of his childhood and I would try and recreate them for him. So, they would be quite odd things like mince cooked in the oven with a layer of beaten egg with kind of gravy. Which again was an approximation of something that he had known as a child, like a congee type thing’.*
(Former carer of parent, female, mixed Chinese/White ethnicity, C1.)

*‘I guess for somebody with dementia, the food and drink, especially familiar food, to bring them back to the phase where they were with their families, as children. So, that reminiscence almost. I think that was really a good thing.*’(Carer of parent, female, Black ethnicity, C8.)

Some carers reported that celebrating cultural and religious festivals also helped to encourage their relatives with dementia to eat and take an interest in food:

*‘Basically, he’s excited on those special [Onam festival] days. Keep him motivated, keep him reminded. And he used to give suggestions, oh, this is how we do the Sadhya [special meal served during the Onam festival], this is how that has to be done, you are not doing the right thing. He can remember some of the things.*’(Former carer of parent-in-law, male, South Asian ethnicity, C11.)

*‘I would say definitely. I’ve noticed that [she eats more], yes. Christmas and Easter she does eat more.*’(Carer of parent, female, Black ethnicity, C8.)

A few carers found that involving their relative with dementia in cooking and preparing meals, particularly if cooking had formed a big part of their predementia identity and if using culturally familiar food preparation methods, helped encourage reminiscence (as well as social interaction), which improved interest and appetite:


*‘We wash the meat before we, not in salt, sometimes with lemons and stuff and then you add the seasoning. So, I might marinate the meat the day before and then cook it the next day. So, the prep before… And so mum’s involved in the prep, you know?’*
(Carer of parent, female, Black ethnicity, C8.)

Many carers reported that eating together as a family, either during family mealtimes or cultural festivals, and particularly when grandchildren were present, promoted social interaction, connectedness and interest in mealtimes, as well as provided social cues for eating and drinking:


*‘I don’t always eat with her but I try to do that more and more because if we’re eating as a family, then she’s more inclined to eat. So, make it a social event.’*
(Carer of parent, female, Black ethnicity, C8.)


*‘I think food is something which we always as a family has brought us together. That’s really an odd thing to say but it is what it is.’*
(Carer of parent, male, South Asian ethnicity, C2.)


*‘Because when she saw the kids, she lit up to some extent, even when she was very, very, very confused. It’s strange that though, when they [grandchildren] would eat, she wouldn’t necessarily eat, but she would have a little drink or something. It was almost like as if she was pleasing them, it’s weird, it’s really weird.’*
(Former carer of parent-in-law, male, Black ethnicity, C4.)

Most participants living with dementia also reported a preference to eat with others, and particularly with family members, rather than alone:


*‘If I’m not alone that’s one thing, you know if I have my family, my grandchildren. They phone and they say they’re coming and so I prepare for them and eat with them and talk with them and that’s very nice.’*
(Person living with dementia, female, North African descent, PD1.)

A minority of carers mentioned that their relative living with dementia was more likely to eat or drink in social environments outside of the home, where there was perhaps less pressure around mealtimes or where different social etiquettes might apply:


*‘And you know, English people, they always serve you with tea and biscuits and then we let her have one or two. But, if the plate is there, she’ll keep on going for it thinking that it’s… She can’t help herself because she’s forgotten that she’s already had one or two and that’s it, whereas Dad will get really cross with her because she keeps reaching for the biscuits.’*
(Carer of parent, female, South Asian ethnicity, C9.)

#### 3.1.3. Barriers to Engaging in Cultural Food Practices

Whilst celebrating cultural traditions and providing culturally familiar foods encouraged many people living with dementia to eat, for others, their diets had evolved to include predominantly Western foods due to the impact of dementia, health conditions or external factors, such as time pressures on carers and generational differences.

Carer strain and generational differences

All carers interviewed reported that managing eating and drinking for their relatives with dementia was a time-consuming and laboured process, and, for some, Western foods (over certain traditional foods) were perceived as easier and less time consuming to prepare, with potentially more ready-meal and convenience options:

*‘With the ready meals, it’s definitely more British. And I try to cook as much as I can, but I often don’t have time to cook a full meal. And mum’s not fussy.*’(Carer of parent, female, Black ethnicity, C12.)

A minority of participants living with dementia, and particularly if they relied on others for their meals, mentioned generational differences in cooking styles and less familiarity with cooking traditional foods as being the reason for their more Westernised diets:

*‘Because she [daughter-in-law] is born here…. and she doesn’t know any Indian dishes, so I do miss those ones.*’(Person living with dementia, male, South Asian ethnicity, PD5.)

Changes in taste and perception

Some people living with dementia reported an altered sense of taste, which affected their usual diets. Some were less likely to eat food perceived as too spicy and therefore avoided certain cultural foods, or required them to be prepared and cooked in a different way:

*‘My traditional West Indian meal, I don’t go for that now anymore. No, very few things I’ll eat, and then I cook it a different way, because West Indians like their food very spicy, and spicy’s out for me. So I’ll cook the same thing, but not the way they… Like frying things. No, I don’t go for fried things. Steam or boil*.’(Person living with dementia, female, Black ethnicity, PD2.)

A few carers reported that their relatives with dementia were more likely than before to try new foods that were not culturally familiar to them (examples included pizza, cheese or sushi), either because their sense of taste had changed, previously held cultural reservations were now forgotten or they were attracted by their appearance, and offering these foods could sometimes encourage them to eat more:


*‘I come from, and my parents come from a culture where food is just put on the plate. Here we go, one big, you know, this, that and the other. Although it is prepared nicely, but it’s not the way that English cuisine is prepared. Especially… they watch a lot of these [cooking] programmes now; I suddenly realised, day-time TV, all of these things prepared beautifully. Suddenly you realise, if you do that, it’s attractive.’*
(Carer of parent, male, South Asian ethnicity, C2.)

Apathy and confusion

Some carers reported that cultural festivals and social interactions could lead to distress and agitation in those with more advanced dementia. These carers subsequently avoided exposing their relatives with dementia to these situations, which meant that they were less likely to celebrate cultural festivals as a family, and that the person with dementia was more likely to eat alone or in the presence of a single carer, rather than share in family or extended family mealtimes:

*‘Festival times, we haven’t had much festival time for the last two years because obviously, he is so confused if I take him out.*’(Carer of spouse, female, South Asian ethnicity, C3.)

*‘But now, she doesn’t remember any festival even if I say to her, it’s Diwali, she’s forgotten. So she doesn’t know. So we offer her and when we are offering her food, to say it’s Diwali so we made this Mum, do you remember? And try to engage her, but nothing goes through.*’(Carer of parent, female, South Asian ethnicity, C9.)

For these carers, upholding a routine for their relative with dementia was seen as preferable to reducing the frequency of distressed behaviour; whilst many acknowledged that this could lead to greater social isolation, most felt that this was preferred by the person with dementia as it was less disruptive to them. However, social isolation and missing significant events and festivals appeared to be felt acutely by the carers:

*‘I won’t even tell him, I mean, it’s our anniversary or it’s a birthday or anything because first of all, he doesn’t understand. Second, then I get depressed myself. Oh, it’s Eid day and I’m at home and I’m not doing anything, so just take it as a normal day because he’s got his routines.*’(Carer of spouse, female, South Asian ethnicity, C3.)

A few carers reported that their relative with dementia, particularly those in the more advanced stages, displayed apathy or a lack of interest and pleasure in food, and that using culturally familiar foods as a strategy to engage them had not worked, even if food had formed a large part of their identity previously:


*‘Participant: Yes, she used to cook a lot. She was a really good cook, and she would cook dishes from her childhood, and Caribbean food. So, she did enjoy trying out different dishes. She definitely loved cooking and eating different foods.*

*Researcher: And does she have that same interest now do you think in food?*

*Participant: Not at all. Completely changed. She definitely can’t cook anymore. She’s just not really interested, unless she’s hungry. She just wants to get rid of the feeling of hunger. It’s not like she’s interested in certain dishes or anything.’*
(Carer of parent, female, Black ethnicity, C12.)


*‘So, Saturday was soup day in Jamaican culture and Sunday was rice and peas and chicken. As she became increasingly worse those things didn’t matter to her anymore.’*
(Former carer of parent-in-law, male, Black ethnicity, C4.)

Using food to take control and manage health

For some carers and people living with dementia, food and drink were perceived as having the power to slow the dementia trajectory, giving some a sense of control over the dementia. For carers, this involved the careful selection, preparation and management of food and drink, whilst, for people living with dementia, this affected their food choices and acceptance of certain foods:

*‘But I realise, if I put that effort in now [in food preparation], I’ll probably save them downward progression [of dementia].*’(Carer of parent, male, South Asian ethnicity, C2.)


*‘I think that food-wise I’m glad. I don’t know if it’s good to say, but I think this illness [dementia] has changed me completely. That’s why I have this faith that I’m going to be better, because I’m doing all the right things and it is it that has changed me to live the way I am.’*
(Person living with dementia, female, Black ethnicity, PD2.)

Many carers and people living with dementia perceived food and drink to be important for general health, as well as for the management of accompanying health conditions, such as diabetes, blood pressure, constipation and urinary incontinence:


*‘It’s not just about food or hunger, it’s about balancing the blood pressure, the diabetes.’*
(Carer of parent, male, South Asian ethnicity, C2.)

For some, trying to balance a healthy diet against eating for pleasure constituted an additional complexity:

*‘I think obviously nutrition should be top. So, you eat the nutritious food before you have the dessert for that reason, but you still have a dessert because you’ve got to have some pleasure as well. Do you know what I mean?*’(Carer of parent, female, Black ethnicity, C8.)

Some carers and people living with dementia had moved away from culturally familiar foods towards a more Western diet, either because they felt that some traditional foods were prepared in an unhealthy way (e.g., fried), or because they were more confident in managing health conditions using Western foods (e.g., due to increased knowledge about the nutritional content of these foods compared to other types of foods):

*‘Their whole food, dietary habits have changed. My dad would only eat like Asian cuisine…and, I recently realised that that was what was causing him the constipation... Then I shifted it to the complete English palate food.*’(Carer of parent, male, South Asian ethnicity, C2.)

#### 3.1.4. The Need for Culturally Sensitive Services

Most carers in our sample were managing food and drink for their relatives with dementia at home, and all identified a need for further information and support from health and social care services that are culturally appropriate. Most carers reported not knowing how to access resources on managing food and drink in dementia, and those who had accessed information felt that it was not particularly relevant to home environments, did not take into account how cultural backgrounds could impact on food choices (e.g., dietary restrictions due to religious beliefs) or felt that the advice that was offered (e.g., on texture modification) focused on British/Western foods and was not easily adaptable to foods from different cultures:

*‘There should be a note to say that somebody from certain cultures might do something differently, or the protocol for people of different religions and cultures*.’(Carer of parent, female, Black ethnicity, C12.)

*‘I think it [support] would be good to be adapted to different cultures because then it’s more diverse, isn’t it? I think, one size doesn’t fit all, do you know what I mean? I do think it should be, yes, definitely.*’(Carer of parent, female, Black ethnicity, C8.)

Another carer felt that a more holistic approach was needed, which takes into account the cultural values and beliefs of the person living with dementia:

*‘I would say a dietician, or a sort of holistic person look at mum and just say, what, how can I change things, because it was about changing her food habits and understanding, because we are vegan. I needed somebody who was vegetarian, vegan too.*’(Carer of parent, male, South Asian ethnicity, C5.)

More diversity within the multidisciplinary team was also mentioned as potentially providing more meaningful support for eating and drinking for ethnic minority groups:


*‘Participant: …Maybe the group could be diverse themselves and then they could make contributions. Do you know what I mean?*

*Researcher: So, the multi-disciplinary team themselves could have more diversity?*

*Participant: Absolutely, that’s the way forward anyway. Yes.’*
(Carer of parent, female, Black ethnicity, C8.)

## 4. Discussion

The findings from our study suggest that cultural backgrounds play an important role in influencing attitudes and experiences related to eating, drinking and care. Food and mealtimes were strongly linked to cultural and personal identity, reminiscence and socialising, all of which encouraged people living with dementia to eat and drink. However, dietary choices were also influenced by carer factors, the dementia process and health conditions. Most carers in our sample were currently caring for their relatives with dementia at home, and all identified a need for more culturally relevant support for managing eating and drinking difficulties.

To the best of our knowledge, this is the first published study that has sought to specifically elicit the views of carers supporting those with advanced dementia and people living with early-stage dementia from ethnic minority groups in the United Kingdom with regard to the management of eating and drinking at home. Studies have mainly focused on long-term care settings, despite the majority of dementia care being undertaken at home [14,27]. These studies found that the provision of culturally familiar foods in care homes fostered a sense of cultural identity and dignity, improved nutritional intake and promoted reminiscence [19,22]. Our findings show that this is applicable to home settings, and most of the people with early-stage dementia that were interviewed confirmed that they wished to continue to be offered culturally familiar foods in the future, if they were unable to verbalise or make clear their preferences.


*Food and drink are linked to identity, cultural expression and emotions*


In our study, eating and drinking were perceived to have strong links with identity. In dementia, identity is threatened as social roles and relationships become altered or lost [32]. Studies suggest that control over food choices encourages a sense of agency, decision making and sense of self in the person with dementia [33], and that the failure to provide culturally appropriate foods in care homes could cause a person with dementia to feel unvalued [34]. The few studies that focus on home environments highlight family mealtimes as being important in promoting cultural values [25] and honouring the identity of the person with dementia [24]. Building on these findings and reflecting previous studies [35], we found that maintaining meaningful food-related roles and identities might encourage interest in food and improve oral intake.

Food was also intrinsically connected to emotions of love between carers and their relatives with dementia, as echoed by Brijnath [23], who found that the feeding of sweet foods in Indian culture, in particular, was linked to displays of love, care and pleasure. However, in our sample, eating for pleasure was carefully balanced against using food to control or manage health conditions, which led to feelings of empowerment in both the carers and the people living with dementia. Thus, food and drink were seen to hold simultaneous multiple meanings that were not only linked to cultural identity, love and pleasure, but were also imbued with a (sometimes unrealistic) sense of power and control over health conditions, including the dementia process.


*Practices related to eating and drinking promote reminiscence*


Reminiscence was highlighted in our findings as having a particularly important role in stimulating people with dementia to eat, and it was encouraged by serving familiar cultural and childhood foods, celebrating traditional festivals and involving the person with dementia in meal preparation, which were in line with their eating habits prior to the onset of dementia. Previous studies [24,25] that explore mealtimes at home also highlight their function in supporting reminiscence by engaging the person with dementia in stories about the past. Some researchers have suggested that reminiscence may reduce distressed behaviour and support people with dementia to eat and drink [36,37]. However, whilst reminiscence therapy has been shown to improve interactions and quality of life for people with dementia [37], its effects have not been shown to be consistent and they appear to vary according to the mode of delivery and the setting [38], perhaps warranting further exploration of its specific application to supporting nutrition and hydration for people living with dementia from ethnic minority groups.


*Multiple factors impact the provision of culturally familiar foods*


One study exploring food and drink in dementia in home environments [25] highlighted generational differences in the perceived importance of cultural foods and traditions, with second- and third-generation immigrants aligning themselves more with mainstream culture and moving away from their parents’ food habits. Whilst this was noted in a minority of our sample, most strived to provide cultural foods that were important to their parents. Culturally familiar foods were widely available during the time of the study, reflecting the multicultural populations in/around London. The reasons for not providing culturally familiar foods were more often due to time constraints and pressures on carers, which led some to opt for convenience foods, or due to uncertainties about the nutritional content of foods from different cultures. Our findings suggest that practitioners supporting carers with managing eating and drinking should consider the impact of time pressures, priorities, cultural values and cooking abilities on food choices, and provide information on nutritional content, if needed, for both convenience foods [15,39,40] and different cultural diets.


*Mealtimes and cultural festivals as a means to stimulate social interaction*


We found that social interactions, for example, during family mealtimes and cultural festivals, and particularly when grandchildren were present, encouraged people living with dementia to eat and drink. Other studies in care homes have found that mealtimes signify a social event [41,42,43], and that the environment in which food is served and eaten is important in stimulating older people to eat [44]. Previous studies focusing on home environments [23,24,25,45] found that mealtimes increased the sense of connectedness between family members and encouraged reminiscence. Our findings support this and further suggest that family mealtimes can reinforce previous social or food-related roles, which can help orientate the person living with dementia and uphold a sense of normality. However, for those with more advanced dementia, celebrating festivals and social events could disrupt routines and worsen the symptoms of dementia. This might increase carers’ social isolation and cultural disengagement, as well as that of the person living with dementia, amplifying carer burden and stress [12,13,46].


*Support for carers is needed to manage nutrition and hydration at home*


Nearly all of the carers that were interviewed felt a sense of cultural duty to care for their relative living with dementia at home, but there was generally a low awareness of the available community support services, which confirms other studies [47,48,49]. There are significant expectations of familial care and a reluctance to use care homes amongst South Asian communities in the United Kingdom [50,51], possibly due to cultural stigma, language barriers and concerns over culturally inappropriate care. An ethnographic study from India [23] highlights the importance of *seva* or the duty of care towards older members of the family, which may also be felt in the United Kingdom, as well as the increased burden of care on female carers, which has also been noted in UK studies [52]. Fewer studies have explored the attitudes amongst Black communities in the United Kingdom towards care homes, but one study found more heterogeneous views [53]. Our study, however, suggests that attitudes to care homes are similar in South Asian and Black communities, with common fears over culturally insensitive care, particularly surrounding the limited provision of cultural foods and not celebrating cultural festivals, as well as fears encompassing aspects of personal care and potential language barriers. This led some carers to move, or to consider moving, their relative living with dementia back to their ‘home’ country as an alternative to a care home, where they felt that the care might be more culturally appropriate and acceptable, which is a novel finding that warrants further exploration to see how common this practice is.

### 4.1. Implications for Theory and Practice

All carers expressed a desire for culturally tailored support with regard to managing eating and drinking in dementia, and specifically that the advice should take into account cultural values, diets and religions, but most had difficulty envisioning exactly what form this support should take and had a low awareness of existing services. Increasing the ethnic diversity of support services by recruiting more people from ethnic minority backgrounds was proposed by one participant, which is an observation that has also been noted in previous studies [49,54], but that needs to be set in the context of the Greater London area, where there is substantial diversity amongst care workers [55].

Our findings suggest that support for carers managing eating and drinking at home for someone with moderate or advanced dementia should take a holistic, multidisciplinary approach. Specialist clinicians (e.g., dieticians, speech and language therapists, GPs, nurses, hospital specialists) and homecare workers providing community support are generally required to take up training that promotes cultural awareness, but there is still a need for more support for cultural sensitivity and respect. Professionals must be mindful of how different cultural values and religious beliefs can affect dietary choices. Furthermore, if necessary, healthcare professionals could provide advice on the nutritional content of cultural foods, and possibly on how to fortify these and help with texture modification.

Encouraging reminiscence through the provision of culturally familiar foods, celebrating traditional festivals and involvement in cultural practices related to cooking and eating in the home were shown to encourage people with moderate–advanced dementia to eat, particularly when food once formed a large part of their identity, and it is strongly encouraged in the current good-practice guidelines [18]. Social interaction around family mealtimes, in particular, or during cultural celebrations, was often positively associated with encouraging eating. Most carers intuitively did these things but including them as potential strategies in a nutritional resource may be helpful (see [56]; this resource was informed by the findings from this study).

However, strategies focusing on cultural or social engagement were not always successful in encouraging people living with dementia to eat, and particularly those with distressed behaviour, which often led to increased carer isolation and burden. Supporting carers with finding alternative strategies for supporting eating and drinking in those with more advanced dementia, and/or providing home care services and respite breaks, may be important for these groups.

Our findings also have implications for care homes and for their marketing and practices, in light of the significant fears expressed by our sample over culturally insensitive care in such settings. 

Thomas Kitwood’s theoretical framework of personhood [57,58] underpins person-centred dementia care in the United Kingdom. Whilst this model has been useful in affecting positive change in the way we care for those living with dementia, it does not take into account structural and societal inequalities, and there is little understanding of how this concept is operationalised on the ground, particularly by and for ethnic minority groups. A recent study exploring the experiences of people with dementia and carers from ethnic minority groups in London during the COVID-19 pandemic highlighted the lack of provision of culturally appropriate foods in home-delivered food parcels, which adversely affected the wellbeing, and sometimes even the nutritional status, of the participants [59]. This study also drew attention to other structural inequalities in dementia care for these groups. Our study findings suggest that an intersectionality approach [60] to understanding the experiences of these groups is vital, as it allows us to take into consideration multiple influencing factors over an individual’s lifetime, such as gender, ethnicity, socioeconomic status and age, and how these combine together to forge unique experiences, attitudes and behaviours, whilst at the same time, providing a lens through which to identify and locate structural inequalities.

Methodologically, our findings highlight that including people from ethnic minority groups in developing resources related to managing eating and drinking in dementia is vital to ensure cultural diversity and the appropriateness of future support [61], and also that awareness needs to be raised amongst these communities of existing services. The findings from this study have directly helped to inform a co-designed information resource for carers managing eating and drinking difficulties at home [56].

### 4.2. Strengths and Limitations

We included participants from diverse socioeconomic neighbourhoods and recruited former and current carers, which enabled us to obtain broad perspectives about caring. However, most carers had completed full-time education, which meant that we may not have elicited the views of those from other sociodemographic groups. Three interviews were conducted remotely, which may not have yielded the same depth of information as face-to-face interviews. Another limitation is the small number of people living with dementia who were interviewed. Unfortunately, despite repeated attempts, we were not able to identify more participants to take part. We utilised the concept of information power to determine overall sample size [29]. The more information power a sample has, the smaller the sample is required to be; our sample had a high level of information power, as our study aims were specific, and our sample included detailed participant cases and displayed a high quality of dialogue [29]. Ethnic minority groups are often difficult to recruit from, which is compounded even further when recruiting from other hard-to-reach groups, such as carers and people living with dementia. Despite this, we assessed that our overall sample had sufficient information power to provide valuable data. This study primarily focused on South Asian and Black ethnic groups, whose views may not be transferable to other minority groups; furthermore, these labels themselves are somewhat reductionist and fail to capture the diverse ethnicities that they represent. One final consideration is that one of the interviewers was a GP (not revealed to participants unless directly asked) from a South Asian background, which may have influenced the participants’ openness during the interviews, both positively (higher levels of trust and relatability) or negatively (less likely to criticise health services/GPs).

## 5. Conclusions

Managing eating and drinking for someone living with dementia in a home environment is challenging, and those from ethnic minority backgrounds do not always receive support that is culturally relevant. Our study sheds light on the cultural importance of food and drink in these groups, and of the positive role that reminiscence and social interaction may have in encouraging people living with dementia to eat and drink. These strategies, however, were often considered inappropriate for those with more advanced dementia. Increasing knowledge of existing services, ensuring that support takes into account cultural values and religious beliefs, and tailoring nutritional advice to culturally familiar foods are tasks for dementia services. Going forward, as we emerge from the shadow of the COVID-19 pandemic, we must ensure that the existing structural inequalities in dementia care for ethnic minority groups are addressed, and this may necessitate transcending or restructuring current personhood theoretical models of dementia [60,62,63], as well as championing more inclusive research methodologies.

## Figures and Tables

**Table 1 nutrients-14-02395-t001:** Inclusion and Exclusion Criteria.

Inclusion Criteria	Exclusion Criteria
Current/former adult carer of someone with moderate to advanced dementia (based on carer’s assessment) living at home	Bereaved within past 3 months (it was felt that research may be intrusive in the context of grief)
*or* Person living with early stage dementia *from* Ethnic minority background (focusing on, but not limited to, South Asian and Black ethnicities)	Carer with cognitive impairment Dementia diagnosed less than 6 months ago (as may not yet have had discussions around potential eating/drinking difficulties, or may still be coming to terms with the diagnosis) Lacking capacity to consent

**Table 2 nutrients-14-02395-t002:** Demographic Characteristics of Participants.

Demographic		Carers (*n* = 12)	PLWD * (*n* = 5)
Age (years)	<40	2	0
	40–49	3	0
	50–59	2	1
	60–69	2	0
	70–79	2	2
	>80	1 Mean: 55.25 years old (range: 29–81)	2 Mean: 75.6 years old (range: 57–87)
Gender	Female	8	3
	Male	4	2
Marital Status	Married	8	2
	Single	3	0
	Divorced	1	1
	Widowed	0	2
Ethnicity	Indian	4	3
	Pakistani	2	0
	Bangladeshi	1	0
	Black Caribbean	3	1
	Chinese	1	0
	Mixed Chinese/White	1	0
	Other (North African)	0	1
Age Left Education	Under 17 years old	0	2
	17–20 years old	3	0
	>20 years old	9	3

* PLWD: people living with dementia.

**Table 3 nutrients-14-02395-t003:** Illustrative Quotes for Themes.

Theme 1: Cultural Expectations around Care	
T1Q1	‘But the South Asian community, when I consider my father-in-law, he always thinks that it’s his wife’s duty [to undertake all caring duties, including preparing meals] and he didn’t need any carers [care workers] to come in and go out, it all has to be done by his wife.’ (Former carer of parent-in-law, male, South Asian ethnicity, C11.)
T1Q2	‘It was unfair on my wife, she had to bear that burden. And I used to try and make my mum understand. Mum, be appreciative of what the fact is, that my wife is putting in this effort, making it [meals] for you. God forbid if you were alone with me, how would you have survived?’ (Carer of parent, male, South Asian ethnicity, C5.)
T1Q3	‘In the Caribbean culture, there’s pressure to keep your parents at home.’ (Carer of parent, female, Black ethnicity, C12.)
T1Q4	‘We’re Asian, so we have to keep our mum at home.’ (Carer of parent-in-law, female, South Asian ethnicity, C6.)
T1Q5	‘I think we’d have 24-h care with carers (care workers) coming in [rather than a care home]… because I think mum would lose the will to live because the food is not going to be culture-specific. It’s not like school, at least schools you take in celebrations, so if there’s Eid or Ramadan you include it. But in care homes, I don’t think they do stuff like that. So, if they did, if they made it more multicultural, it would be more bearable.’ (Carer of parent, female, Black ethnicity, C8.)
T1Q6	‘Talking about food, my mum’s carer (care worker), she mentioned about this chap. He was, I think he was Greek Cypriot. And she went to visit somebody else in this care home, and they had given him some sausage and mash or something. He looked at it and he threw it across the room, and, of course, they’re going to say that he’s aggressive. He wasn’t aggressive, he was confused, he was frustrated and the food wasn’t right……He probably didn’t recognise that it was food. He wasn’t being horrible, he was just was like, what the hell are you giving me?’ (Carer of parent, female, Black ethnicity, C8.)
T1Q7	‘I used to work for the [name of charity] and I see people brought their family from Asia into England at 80, they come from a village and in no time they deteriorated. They deteriorated because they moved them out of their cultural place and they come here, because families think, they’re going to have a better lifestyle in terms of food, drink, whatever and a better environment. But you take that person out of that culture and bring them here, this place is isolating, but if you live in a small village in Kolkata or something it’s fantastic, do you know what I mean?’ (Former carer of parent-in-law, male, South Asian ethnicity, C11.)
**Theme 2: Food as an expression of culture, identity and relationships**	
*Identity, roles and emotions*	
T2Q1	‘It’s more about being an Asian woman. Right from the early age, it’s like looking after the family. So, she’s always, whoever comes, a guest in the house, she will always offer them food.,,,,[..]……. Even now, today, the lady [cleaner] was saying, the new lady that’s she’s got. She said, are you still cooking? Oh yes, when I’m home I still cook. So she still thinks she does the cooking.’ (Carer of parent, female, South Asian ethnicity, C9.)
T2Q2	‘She will not, by herself, will not come to the table…. And I’ll say come on, come to the table. But the one thing that she still does is she always looks at my dad to say, are you not joining us? Come on, sort of a thing, you know. It’s that her… Because she, being the woman, looked after the family for all her life.’ (Carer of parent, female, South Asian ethnicity, C9.)
T2Q3	‘He’s the main person, the head of the family. He’ll sit on one side in the bigger chair and he wants everyone to be with him [during meal-times]’. (Former carer of parent-in-law, male, South Asian ethnicity, C11.)
T2Q4	‘I start noticing that when she cooked the soup on a Saturday, that’s when it all came about, it didn’t taste the same. I remember saying, here, Mother, why this soup aren’t taste nothing. She used to get quite insulted about it, she used to get uptight…. And then we started thinking, something’s wrong here.’ (Former carer of parent-in-law, male, Black ethnicity, C4.)
T2Q5	‘Researcher: And what’s important to you about eating? Participant: …Teaching them how to cook my way. Researcher: Are there specific recipes that are important for you to teach? Participant: Yes…For instance stuffed onions which takes a lot of time. I taught them but they love doing it so I don’t do it anymore, they do it. And things like that, yes.’ (Person living with dementia, female, North African descent, PD1.)
T2Q6	‘And I’m preparing the food, getting home quickly, putting it on to the table, putting it in front of them. You forget, although you’ve done all of that, the person’s still there. In their mind, they would have seen you do all of that.’ (Carer of parent, male, South Asian ethnicity, C2.)
T2Q7	‘So I used to, I’ve got a sweet tooth, so I used to sometimes say, I’m having a bit of chocolate or something, and I used to see her noticing me and I felt bad, because I didn’t want her to feel he’s not giving me these things. So I used to give her.’ (Carer of parent, male, South Asian ethnicity, C5.)
T2Q8	‘To be honest with you, for me it was a huge learning curve going on and there was a role reversal going on, which I had heard about but never experienced.’ (Carer of parent, male, South Asian ethnicity, C2.)
T2Q9	‘It was sad to see how this joyful woman who liked her food detested food.’ (Former carer of parent-in-law, male, Black ethnicity, C4.)
T2Q10	‘I wouldn’t like what they [the hospital staff] give me to eat, would I? My daughter would have to bring things to me every day, and I thought she wouldn’t be able to do it, because she would break down. Because when I was in hospital and I wouldn’t eat it, I could see it on her face.’ (Person living with dementia, female, Black ethnicity, PD2.)
*Cultural and religious beliefs*	
T2Q11	‘We insisted on getting things that she liked and things that were cultural to her.’ (Former carer of parent-in-law, male, Black ethnicity, C4.)
T2Q12	‘I was born in [North African Country] and I’m Jewish. And so there are specific Jewish foods and my family follow which I carry on [making] here.’ (Person living with dementia, female, North African descent, PD1.)
T2Q13	‘I think it’s quite difficult to go to a completely different place and have completely different food, in terms of if you’re Asian and then just English food. I think it’s important that they have things that they’re used to.’ (Person living with dementia, female, South Asian ethnicity, PD4.)
T2Q14	‘We know for a fact that a lot of different people, with different religious beliefs will not eat certain foods, and so on. My mum is a prime example.’ (Carer of parent, male, South Asian ethnicity, C5–Hindu vegetarian.)
T2Q15	‘Yes, garlic, onion, like the cassava that grows under the ground, and the sweet potatoes, we don’t eat’ (carer of spouse, female, South Asian ethnicity, C10–Jain religion).
T2Q16	‘She went to this Ayurveda doctor, and it was herbal medication he was giving, and changing your diet a bit, and that fixed the problem for her. So that was brilliant.’ (Carer of parent, male, South Asian ethnicity, C5.)
T2Q17	‘She’s always been vegetarian, yes. So that bit… Because one day, at Dad’s house, by mistake, she did help herself from the fridge, which…...She helped herself with a non-veg dish and she didn’t realise it. It was only later on that Dad realised that she had eaten it.’ (Carer of parent, female, South Asian ethnicity, C9.)
*Food as a connection to the past and others: reminiscence and social interaction*	
T2Q18	‘And when he was very frail there was a time when he did crave the things of his childhood and I would try and recreate them for him. So, they would be quite odd things like mince cooked in the oven with a layer of beaten egg with kind of gravy. Which again was an approximation of something that he had known as a child, like a congee type thing’ (Former carer of parent, female, mixed Chinese/White ethnicity, C1).
T2Q19	‘I guess for somebody with dementia, the food and drink, especially familiar food, to bring them back to the phase where they were with their families, as children. So, that reminiscence almost. I think that was really a good thing.’ (Carer of parent, female, Black ethnicity, C8.)
T2Q20	‘Basically, he’s excited on those special [Onam festival] days. Keep him motivated, keep him reminded. And he used to give suggestions, oh, this is how we do the Sadhya [special meal served during the Onam festival], this is how that has to be done, you are not doing the right thing. He can remember some of the things.’ (Former carer of parent-in-law, male, South Asian ethnicity, C11.)
T2Q21	‘I would say definitely. I’ve noticed that [she eats more], yes. Christmas and Easter she does eat more.’ (Carer of parent, female, Black ethnicity, C8.)
T2Q22	‘We wash the meat before we, not in salt, sometimes with lemons and stuff and then you add the seasoning. So, I might marinate the meat the day before and then cook it the next day. So, the prep before… And so mum’s involved in the prep, you know?’ (Carer of parent, female, Black ethnicity, C8.)
T2Q23	‘I don’t always eat with her but I try to do that more and more because if we’re eating as a family, then she’s more inclined to eat. So, make it a social event.’ (Carer of parent, female, Black ethnicity, C8.)
T2Q24	‘I think food is something which we always as a family has brought us together. That’s really an odd thing to say but it is what it is.’ (Carer of parent, male, South Asian ethnicity, C2.)
T2Q25	‘Because when she saw the kids, she lit up to some extent, even when she was very, very, very confused. It’s strange that though, when they [grandchildren] would eat, she wouldn’t necessarily eat, but she would have a little drink or something. It was almost like as if she was pleasing them, it’s weird, it’s really weird.’ (Former carer of parent-in-law, male, Black ethnicity, C4.)
T2Q26	‘If I’m not alone that’s one thing, you know if I have my family, my grandchildren. They phone and they say they’re coming and so I prepare for them and eat with them and talk with them and that’s very nice.’ (Person living with dementia, female, North African descent, PD1.)
T2Q27	‘And you know, English people, they always serve you with tea and biscuits and then we let her have one or two. But, if the plate is there, she’ll keep on going for it thinking that it’s… She can’t help herself because she’s forgotten that she’s already had one or two and that’s it, whereas Dad will get really cross with her because she keeps reaching for the biscuits.’ (Carer of parent, female, South Asian ethnicity, C9.)
**Theme 3: Barriers to engaging in cultural food practices**	
*Carer strain and generational differences*	
T3Q1	‘With the ready meals, it’s definitely more British. And I try to cook as much as I can, but I often don’t have time to cook a full meal. And mum’s not fussy.’ (Carer of parent, female, Black ethnicity, C12.)
T3Q2	‘Because she [daughter-in-law] is born here…. and she doesn’t know any Indian dishes, so I do miss those ones.’ (Person living with dementia, male, South Asian ethnicity, PD5.)
*Changes in taste and perception*	
T3Q3	‘My traditional West Indian meal, I don’t go for that now anymore. No, very few things I’ll eat, and then I cook it a different way, because West Indians like their food very spicy, and spicy’s out for me. So I’ll cook the same thing, but not the way they… Like frying things. No, I don’t go for fried things. Steam or boil.’ (Person living with dementia, female, Black ethnicity, PD2.)
T3Q4	‘I come from, and my parents come from a culture where food is just put on the plate. Here we go, one big, you know, this, that and the other. Although it is prepared nicely, but it’s not the way that English cuisine is prepared. Especially… they watch a lot of these [cooking] programmes now; I suddenly realised, day-time TV, all of these things prepared beautifully. Suddenly you realise, if you do that, it’s attractive.’ (Carer of parent, male, South Asian ethnicity, C2.)
*Apathy and confusion*	
T3Q5	‘Festival times, we haven’t had much festival time for the last two years because obviously, he is so confused if I take him out.’ (Carer of spouse, female, South Asian ethnicity, C3.)
T3Q6	‘But now, she doesn’t remember any festival even if I say to her, it’s Diwali, she’s forgotten. So she doesn’t know. So we offer her and when we are offering her food, to say it’s Diwali so we made this Mum, do you remember? And try to engage her, but nothing goes through.’ (Carer of parent, female, South Asian ethnicity, C9.)
T3Q7	‘I won’t even tell him, I mean, it’s our anniversary or it’s a birthday or anything because first of all, he doesn’t understand. Second, then I get depressed myself. Oh, it’s Eid day and I’m at home and I’m not doing anything, so just take it as a normal day because he’s got his routines.’ (Carer of spouse, female, South Asian ethnicity, C3.)
T3Q8	‘Participant: Yes, she used to cook a lot. She was a really good cook, and she would cook dishes from her childhood, and Caribbean food. So, she did enjoy trying out different dishes. She definitely loved cooking and eating different foods. Researcher: And does she have that same interest now do you think in food? Participant: Not at all. Completely changed. She definitely can’t cook anymore. She’s just not really interested, unless she’s hungry. She just wants to get rid of the feeling of hunger. It’s not like she’s interested in certain dishes or anything.’ (Carer of parent, female, Black ethnicity, C12.)
T3Q9	‘So, Saturday was soup day in Jamaican culture and Sunday was rice and peas and chicken. As she became increasingly worse those things didn’t matter to her anymore.’ (Former carer of parent-in-law, male, Black ethnicity, C4.)
*Using food to take control and manage health*	
T3Q10	‘But I realise, if I put that effort in now [in food preparation], I’ll probably save them downward progression [of dementia].’ (Carer of parent, male, South Asian ethnicity, C2.)
T3Q11	‘I think that food-wise I’m glad. I don’t know if it’s good to say, but I think this illness [dementia] has changed me completely. That’s why I have this faith that I’m going to be better, because I’m doing all the right things and it is it that has changed me to live the way I am.’ (Person living with dementia, female, Black ethnicity, PD2.)
T2Q12	‘It’s not just about food or hunger, it’s about balancing the blood pressure, the diabetes.’ (Carer of parent, male, South Asian ethnicity, C2.)
T3Q13	‘I think obviously nutrition should be top. So, you eat the nutritious food before you have the dessert for that reason, but you still have a dessert because you’ve got to have some pleasure as well. Do you know what I mean?’ (Carer of parent, female, Black ethnicity, C8.)
T3Q14	‘Their whole food, dietary habits have changed. My dad would only eat like Asian cuisine…and, I recently realised that that was what was causing him the constipation... Then I shifted it to the complete English palate food.’ (Carer of parent, male, South Asian ethnicity, C2.)
**Theme 4: The need for culturally sensitive services**	
T4Q1	‘There should be a note to say that somebody from certain cultures might do something differently, or the protocol for people of different religions and cultures.’ (Carer of parent, female, Black ethnicity, C12.)
T4Q2	‘I think it [support] would be good to be adapted to different cultures because then it’s more diverse, isn’t it? I think, one size doesn’t fit all, do you know what I mean? I do think it should be, yes, definitely.’ (Carer of parent, female, Black ethnicity, C8.)
T4Q4	‘I would say a dietician, or a sort of holistic person look at mum and just say, what, how can I change things, because it was about changing her food habits and understanding, because we are vegan. I needed somebody who was vegetarian, vegan too.’ (Carer of parent, male, South Asian ethnicity, C5.)
T4Q5	‘Participant: …Maybe the group could be diverse themselves and then they could make contributions. Do you know what I mean?’ Researcher: So, the multi-disciplinary team themselves could have more diversity? Participant: Absolutely, that’s the way forward anyway. Yes.’ (Carer of parent, female, Black ethnicity, C8.)

## Data Availability

This is a qualitative study and, therefore, the data generated are not suitable for sharing beyond that contained within the report. Further information can be obtained from the corresponding author.

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
