# Peer review of "Experiences of Carers and People with Dementia from Ethnic Minority Groups Managing Eating and Drinking at Home in the United Kingdom"

_nutrients, 2022, doi:10.3390/nu14122395_

Round 1
Reviewer 1 Report
Nair and colleagues in the present study entitled ‘Experiences of carers and people with dementia from ethnic minority groups managing eating and drinking at home in the UK: a qualitative study’, investigated the current status of knowledge of the experiences of people with dementia and carers from ethnic minority groups in England in relation to managing to eat and drinking at home. For this purpose, they undertook qualitative interviews from 17 carers and people with dementia to explore the meaning of food, the impact of dementia on eating and drinking and carers’ experiences of support. Results from interviews highlighted how providing culturally familiar foods, celebrating traditional festivals and supporting previous food-related roles promoted reminiscence in patients with AD, which encouraged them to eat and drink, as did social interactions. However, these strategies sometimes led to distress in those with more advanced dementia. The authors concluded by stating that carers expressed a need for culturally tailored support for managing dementia-related eating and drinking difficulties at home, and how healthcare professionals must provide contextually relevant advice to carers, explaining how cultural backgrounds can affect dietary choices.
The main strength of this manuscript is that it addresses an interesting and timely question, providing a captivating interpretation and describing how carers supporting those with advanced dementia and people living with early-stage dementia from ethnic minority groups in the UK, regarding the management of eating and drinking at home. In general, I think the idea of this perspective article is really interesting and the authors’ fascinating observations on this timely topic may be of interest to the readers of Nutrients. However, some comments, as well as some crucial evidence that should be included to support the author’s argumentation, needed to be addressed to improve the quality of the manuscript, its adequacy, and its readability prior to the publication in the present form, in particular reshaping parts of the Introduction and Discussion sections.
Please consider the following comments:
- Abstract: Please proportionally present background, purpose, methods, results, and conclusion.
- In general, I recommend authors to use more evidence to back their claims, especially in the Introduction of the article, which I believe is currently lacking. Thus, I recommend the authors to attempt to deepen the subject of their manuscript, as the bibliography is too concise: nonetheless, in my opinion, less than 60/70 articles for a research paper are really insufficient. Indeed, currently, authors cite only 50 papers, and they are too low. Therefore, I suggest the authors to focus their efforts on researching more relevant literature: I believe that adding more studies and reviews will help them to provide better and more accurate background to this study. In this review, I will try to help the authors by suggesting some relevant literature of my knowledge that suits their manuscript.
- Introduction: I suggest the authors to reshape the Introduction section, which seems not enough extensive and it does not seem to consider, in most cases, all the available studies in the literature that have acknowledged information about pathogenesis, diagnostics, and therapeutics of Alzheimer’s disease. In my opinion, this section is quite thin and dispersive, and I believe that information about the pathophysiology of Alzheimer’s disease is completely missing, specifically its definition, causes, symptoms and related neurocognitive changes. Considering that this study's main focus is to deepen current understanding of Alzheimer's disease pathogenesis and risk factors, I suggest the authors to make such effort to provide a brief overview of the pertinent published literature that offers a perspective on pathophysiology and neurological changes of AD, because as it stands, there is no mention of this in the manuscript. To this end, to gain a more comprehensive and appropriate theoretical background on this topic, I would recommend citing a review that examined pathophysiological basis and biomarkers of AD pathology (https://doi.org/10.3390/ijms21249338) and a study in which authors investigated age-related impairments in the ability to process contextual information and in the regulation of responses to threat, addressing that structural and physiological alterations in the prefrontal cortex and medial temporal lobe determine cognitive changes in advanced aging, that can eventually cause patterns of cognitive dysfunctions observed in patients with AD/MCI (https://doi.org/10.1038/s41598-018-31000-9).
- Introduction: In according with the previous point raised, I would also suggest adding information from recent evidence that has focused on cognitive symptoms (i.e., dysfunction in attention and emotion perception) in neurologic and brain-damaged patients, and highlighted the role that specific dysfunctional brain regions, such as amygdala and superior temporal sulcus (STS), have on recognition and identification of non-verbal communicative signals of emotion in those suffering from Alzheimer's disease (https://doi.org/10.3390/biomedicines10030627). I believe that this evidence will help to provide a more coherent and defined background highlighting emotional processing in patients with Alzheimer's disease.
- The objectives of this study are generally clear and to the point; however, I believe that there are some ambiguous points that require clarification or refining. I think that authors here need to be explicit regarding how they operationally assessed participants’ ethnicity group and how they determined the relationship between ethnicity and attitude related to eating, drinking and care.
- Participants: The sample size is too small. This may reduce the power of the study, therefore I suggest to report a power analysis that will determine the sample size that is most suitable to gain level of significance.
- Inclusion/Exclusion criteria and Data collection: I suggest the Authors to reorganize/rewrite these paragraphs because, as it stands, these sections are way too much inhomogeneous and dispersive, and describe the research procedures in an excessively broad way.
- Results: In my opinion, this section is well organized, but it illustrates findings in an excessively broad way, without really providing full statistical details, to ensure in-depth understanding and replicability of the findings. Indeed, I believe that it is necessary for the authors to present their findings with a precise description not just in the main text, but also in descriptive tables.
- Discussion: In this final section, the authors described the results and their argumentation and captured the state of the art well; however, I would have liked to see some views on a way forward. I believe that the authors should make an effort, trying to explain the theoretical implication as well as the translational application of this research article, to adequately convey what they believe is the take-home message of their study. Discussion of theoretical and methodological avenues in need of refinement is necessary, as well as suggestions of a path forward in the understanding of supportive interventions and treatment for people with dementia. In this regard, recent evidence suggests that the application of new methods in neurodegenerative disorders’ treatment, such as the Non-invasive brain stimulation techniques (NIBS), have shown promising results in humans (https://doi.org/10.1016/j.arr.2021.101499). Importantly, I recommend referring recent studies that revealed that the application of NIBS induces long-lasting effects, noninvasively modulating the cortical excitability, and modulates a variety of cognitive functions: for example, a recent review acknowledged the implementation of NIBS to modulate in general emotional memories (https://doi.org/10.1016/j.neubiorev.2021.04.036). Additionally, I would suggest another recent study that illustrated the therapeutic potential of NIBS as a valid alternative for those patients not responding or drug treatments (https://doi.org/10.1016/j.jad.2021.02.076). In addition to the previously mentioned literature, authors might also see these additional studies that have focused on the efficacy of NIBS and IBS (https://doi.org/10.3389/fpsyt.2018.00201; https://doi.org/10.3389/fnagi.2020.578339) in AD treatment.
- I think the ‘Conclusions’ paragraph would benefit from some thoughts as well as in-depth considerations by the authors because as it stands, it is very descriptive but not enough theoretical as a discussion should be. Authors should make an effort, trying to explain the theoretical implication as well as the translational application of their research.
- In according to the previous comment, I would ask the authors to include a ‘Limitations and future directions’ section before the end of the manuscript, in which authors can describe in detail and report all the technical issues brought to the surface.
- Tables: Please consider including an explanatory caption and all statistical values (such as mean and standard deviation) for each variable.
- References: According to the Journal’s guidelines, authors should have provided the DOI number for each reference.
- According to the Journal’s guidelines, provide only the last name and first name’s initials of individuals authors in the ‘Author Contributions’ section.
Overall, the manuscript contains 1 table and 50 references. In my opinion, the number of references is dramatically low for an original research article, and this prevents the possibility of publishing it in this form. References should be more than 60/70 for original research articles. However, the manuscript might carry important value presenting how carers support those with advanced dementia and people living with early-stage dementia from ethnic minority groups in the UK, regarding the management of eating and drinking at home.
I hope that, after these careful revisions.
I am available for a new round of revision of this article.
Best regards,
Reviewer
Reviewer 2 Report
The authors have studied the eating and drinking habit of people with dementia. Various stages of dementia were chosen, and the participants were also selected randomly. Furthermore, they explored the views related to the cultural importance of food and drink and carers' support and experience. However,
- The number of participants selected is meager for this study. A larger sample group is required.
- The participants' age, general eating habits before dementia, and family history could also affect the results. Authors should add this to their discussion.
- Did the food availability and the native staple diet affect the end result? Please discuss in elaborate.
Round 2
Reviewer 1 Report
I am very pleased to see that the Authors have welcomed my suggestions and have clarified most of the questions I raised in my first round of this review. I believe that this research article addresses an interesting and timely question, exploring the meaning of food, the impact of dementia on eating and drinking of carers’ supporting those with advanced dementia in the UK.